# Low-Level Tolerance to Antibiotic Trimethoprim in QAC-Adapted Subpopulations of *Listeria monocytogenes*

**DOI:** 10.3390/foods10081800

**Published:** 2021-08-04

**Authors:** Divya Kode, Ramakrishna Nannapaneni, Sam Chang

**Affiliations:** Department of Food Science, Nutrition and Health Promotion, Mississippi State University, Mississippi, MS 39762, USA; dsk125@msstate.edu (D.K.); schang@fsnhp.msstate.edu (S.C.)

**Keywords:** *Listeria monocytogenes*, QAC, sublethal adaptation, biocides, trimethoprim, antibiotics

## Abstract

Between January and July 2021, there were as many as 30 recalls in the U.S. due to potential *Listeria monocytogenes* contamination from a variety of food products including muffins, kimchi, chicken salad, ready-to-eat chicken, smoked fish, mushrooms, queso fresco cheese, ice cream, turkey sandwiches, squash, and other foods. A contaminated food chain can serve as a potential vehicle for transmitting antibiotic resistant bacteria since there is a slow emergence of multi-drug antibiotic resistance in *L. monocytogenes*. Biocides are essential for safe food processing, but they may also induce unintended selective pressure at sublethal doses for the expression of antibiotic resistance in *L. monocytogenes*. To better understand the sources of such slow emergence of antibiotic resistance through biocide residues present in the food environments, we are working on the role of sublethal doses of commonly used biocides in defined broth and water models for understanding *L. monocytogenes* adaptation. We recently published the development of low-level tolerance to fluoroquinolone antibiotic ciprofloxacin in quaternary ammonium compound (QAC) adapted subpopulations of *L. monocytogenes* (Microorganisms 9, 1052). Of the six different antibiotics tested to determine heterologous stress adaptation in eight strains of *L. monocytogenes*, trimethoprim was the second one that exhibited low-level tolerance development after continuous exposure (by three approaches) to sublethal concentrations of QAC against actively growing planktonic cells of *L. monocytogenes*. When adapted to daily cycles of fixed or gradually increasing sublethal concentrations of QAC, we observed three main findings in eight *L. monocytogenes* strains against trimethoprim: (a) 3 of the 8 strains exhibited significant increase in short-range minimum inhibitory concentration (MIC) of trimethoprim by 1.7 to 2.5 fold in QAC-adapted subpopulations compared to non-adapted cells (*p* < 0.05); (b) 2 of the 8 strains exhibited significant increase in growth rate in trimethoprim (optical density (OD) by 600 nm at 12 h) by 1.4 to 4.8 fold in QAC-adapted subpopulations compared to non-adapted cells (*p* < 0.05); and (c) 5 of the 8 strains yielded significantly higher survival by 1.3-to-3.1 log CFU/mL in trimethoprim in QAC-adapted subpopulations compared to the non-adapted control (*p* < 0.05). However, for 3/8 strains of *L. monocytogenes*, there was no increase in the survival of QAC-adapted subpopulations compared to non-adapted control in trimethoprim. These findings suggest the potential formation of low-level trimethoprim tolerant subpopulations in some *L. monocytogenes* strains where QAC may be used widely. These experimental models are useful in developing early detection methods for tracking the slow emergence of antibiotic tolerant strains through food chain. Also, these findings are useful in understanding the predisposing conditions leading to slow emergence of antibiotic resistant strains of *L. monocytogenes* in various food production and food processing environments.

## 1. Introduction

Quaternary ammonium compounds (QACs) are a major class of cationic surfactants extensively used in health care industries and in food processing plants due to their low toxicity and non-corrosive properties. Due to their structural capability, various formulations of QACs are available and have shown broad spectrum activity against bacteria, fungi, and protozoa [1]. QACs applied at lethal concentrations are highly effective to maintain food safety and hygiene in the food processing environments. Generally, QACs are used in concentrations ranging from 200–400 ppm on food-contact surfaces. There are some formulations of QACs that are also used at 1000 ppm concentrations [2]. However, it is possible that in certain circumstances the bacteria are exposed to concentrations lower than the killing concentrations. Although QACs are used in concentrations well above the minimum bactericidal concentration (MBC), sublethal concentrations of QAC are frequently formed because of an improper distribution of disinfectants, or by the dilution of disinfectants due to application on wet surfaces, or by the presence of organic matter [3,4]. Complex food processing equipment designs may leave residual organic matter behind which interferes with the disinfectant efficiency [5]. Most QACs are aerobically biodegradable and therefore their concentrations in the environment often fluctuate resulting in the formation of concentration gradients [6].

*Listeria monocytogenes* is an intracellular foodborne pathogen of high-risk public health concern. This pathogen has been isolated from various ready-to-eat food products and from food-contact processing environment such as chillers, conveyors, slicers, packaging machine, and freezers. To combat this pathogen, U.S. regulatory agencies established a ‘zero-tolerance policy’ against *L. monocytogenes* in ready-to- eat foods. Recently, in July 2021, 8.5 million pounds of ready-to-eat chicken products (frozen, fully cooked) were recalled by USDA Food Safety and Inspection Service because of contamination with *L. monocytogenes* after three illnesses and one death were reported [7]. Also, another recent listeriosis outbreak in the U.S. in May 2021 was associated with the consumption of queso fresco cheese where there were twelve hospitalization and one death [8]. In addition to these two recent outbreaks, between January to July 2021, FDA reported 30 food safety recalls in the U.S. due to potential *L. monocytogenes* contamination in a variety of food products including muffins, kimchi, chicken salad, ready-to-eat chicken, smoked fish, mushrooms, queso fresco cheese, ice cream, turkey sandwiches, squash, and other foods [9]. Besides U.S., European data also show that there is in increase in the number of confirmed listeriosis cases in the European Union [10].

Persistent strains of *L. monocytogenes* have been repeatedly isolated from food processing environments after sanitation steps involving QAC. Persisters are subpopulation of cells that are phenotypically diverse and tolerant to various stresses including antibiotics [11]. Persistent *L. monocytogenes* isolates from food processing plants and ecosystems exhibited resistance to QAC [12]. Some isolates of *L. monocytogenes* from food, food environments, animals, and humans from Swiss and Finland exhibited highest prevalence of QAC resistant strains [13]. Concerns have been raised regarding the selection of antibiotic tolerant bacteria after exposure to biocides such as QACs. Exposure to progressively increasing concentrations to QAC in *L. monocytogenes* strains demonstrated increased resistance to cefotaxime, cephalothin and ciprofloxacin [14].

*L. monocytogenes* is naturally susceptible to the clinically relevant antibiotics. Trimethoprim is used in combination with sulfonamides in patients exhibiting an allergic reaction to penicillin, an antibiotic routinely used for listeriosis treatment [15]. Recently, one *L. monocytogenes* isolate from a clinical sample exhibited a high-level resistance against trimethoprim due to the presence of a gene encoding resistance enzyme [16]. *L. monocytogenes* strains exhibiting trimethoprim tolerance have been later isolated from food processing environments [17]. Antibiotic susceptibility tests conducted on 167 *L. monocytogenes* isolates recovered from retail foods such as ready-to-eat (RTE) meats, raw chicken carcass, organically and conventionally grown produce in Florida and Washington showed a reduced susceptibility to trimethoprim and three other antibiotics [18]. Although *L. monocytogenes* strains exhibiting trimethoprim tolerance have been isolated occasionally from food processing environments, there is limited knowledge on the role of QAC in the development of heterologous cross-tolerance response to trimethoprim in *L. monocytogenes*. Therefore, in the present study, we investigated the occurrence of tolerance to trimethoprim among *L. monocytogenes* strains after sublethal exposure to QAC with the following objectives: (1) determine changes in short-range MIC of trimethoprim in QAC-adapted subpopulations of *L. monocytogenes* strains; (2) determine the growth kinetics of the QAC adapted subpopulations of *L. monocytogenes* in trimethoprim-containing broth; and (3) determine the survival of QAC-adapted subpopulations of *L. monocytogenes* in trimethoprim-containing agar. These data are helpful in understanding the potential link between biocide tolerance and emergence of trimethoprim resistant of strains of *L. monocytogenes* that may lead to food safety risk.

## 2. Materials and Methods

### 2.1. Listeria Monocytogenes Strains and Growth Conditions

The eight *L. monocytogenes* strains used in this study are listed in Table 1*. L. monocytogenes* strains supplemented with 25% glycerol were stored at −20 °C in Tryptic soy broth containing 0.6% yeast extract (TSBYE) (BD Bio sciences, San Jose, CA, USA). Working stocks were prepared in 10 mL TSBYE by inoculating a single colony from PALCAM agar plates and incubating at 37 °C for 24 h. Prior to each experiment, broth subcultures were prepared by inoculating a loopful from working stock in 10 mL TSBYE and incubated at 37 °C for 18–24 h to yield a cell suspension of approximately 10^9^ CFU/mL.

### 2.2. Preparation of QAC Solutions

Stock solutions of 5000 µg/mL were prepared by diluting the 50% benzalkonium chloride solution, a first generation QAC (ACROS Organics™, Fair Lawn, NJ, USA) in sterile distilled water. The working stocks of 4 µg/mL and 1 to 8 µg/mL were prepared in TSBYE and 1 to 8 µg/mL was prepared in distilled water prior to each experiment.

### 2.3. L. monocytogenes Adaptation to Sublethal QAC

Planktonic cells of *L. monocytogenes* strains were adapted to fixed or progressively increasing sublethal concentrations of QAC in 24-well plates (Techno Plastic Products, AG, Switzerland) using two broth models and one water model: (1) fixed concentration of QAC in TSBYE (QAC-P1); (2) gradually increasing concentration of QAC in TSBYE (QAC-P2); and (3) gradually increasing concentration of QAC in distilled water (QAC-P3).

### 2.4. Preparation of QAC-Adapted Subpopulation 1 in Broth Model (QAC-P1)

Eight *L. monocytogenes* strains were exposed to a daily fixed concentration of 2 µg/mL sublethal QAC for 5 days in broth model. In a 24-well microtiter an aliquot of 100 µL overnight grown cells diluted to 10^8^ CFU/mL was added to 900 µL of TSBYE to reach 10^7^ CFU/mL final cell concentration. Then, the cells were adapted by adding 200 µL volume of TSBYE containing 4 µg/mL QAC into each well, every hour for up to 5 h to reach 2 µg/mL final QAC concentration in 2 mL/well final volume. The 24-well plate was incubated for 19 h at room temperature (22 °C) until cell concentration reached 10^9^ CFU/mL or max optical density (OD) at 600 nm (>0.9). This process was repeated daily using newly adapted cells for 5 daily cycles (or 5 days) in the same QAC concentration.

### 2.5. Preparation of QAC-Adapted Subpopulation 2 in Broth Model (QAC-P2)

Eight strains of *L. monocytogenes* were adapted to a progressively increasing concentration of QAC with daily increment of 0.5 µg/mL from 0.5 µg/mL to 4 µg/mL for 8 days in broth model. An aliquot of 100 µL of overnight grown cultures (10^9^ CFU/mL) was diluted and added to 900 µL of TSBYE to yield 10^7^ CFU/mL in sterile 24-well polystyrene plates (1 mL/well). Then, 200 µL aliquots of TSBYE containing QAC was added at hourly intervals for 5 h to give a final QAC concentration of 0.5 µg/mL (day 1), 1 µg/mL (day 2), 1.5 µg/mL (day 3), 2.0 µg/mL (day 4), 2.5 µg/mL (day 5), 3 µg/mL (day 6), 3.5 µg/mL (day 7) and 4 µg/mL (day 8) in a total volume of 2 mL/well. The 24-well plate was incubated for 19 h at room temperature (22 °C) until reached 10^9^ CFU/mL or max. OD_600_ (>0.9) and following bacterial growth the cells were challenged with a daily increment of 0.5 µg/mL QAC concentration as described above.

### 2.6. Preparation of QAC-Adapted Subpopulation 3 in Water Model (QAC-P3)

Eight *L. monocytogenes* strains were exposed to gradually increasing concentrations of QAC in 0.5 to 4 µg/mL at 0.5 µg/mL daily increments in water model for 8 days. Overnight grown cells were diluted to 10^8^ CFU/mL in sterile distilled water and an aliquot of 100 µL was inoculated with 100 µL of distilled water containing QAC and incubated for 1 h. These steps were repeated for 8 consecutive days with 0.5 µg/mL increments each day such that the final concentration was 0.5 µg/mL (day 1), 1 µg/mL (day 2), 1.5 µg/mL (day 3), 2.0 µg/mL (day 4), 2.5 µg/mL (day 5), 3 µg/mL (day 6), 3.5 µg/mL (day 7), and 4 µg/mL (day 8). After 1 h exposure the cells were then mixed with TSBYE containing 0.5 µg/mL QAC and incubated at room temperature (22 °C) for 23 h until reached 10^9^ CFU/mL or max OD_600_ (> 0.9). In total, eight consecutive 24 h cultures in 0.5 µg/mL increments of QAC concentrations were performed for each strain.

### 2.7. Preparation of Trimethoprim Solutions

The stock solution for trimethoprim (Trimethoprim, 98%, ACROS Organics™, Fair Lawn, NJ, USA) was prepared at 500 µg/mL stock solution by dissolving in dimethyl sulfoxide (DMSO) and sterilized distilled water (1:20) and stored at 4 °C for 2 weeks. Prior to each experiment, working stock solutions in 0.125 to 1.5 µg/mL were prepared in sterile TSBYE from the stock solution.

### 2.8. Determination of Short-Range MIC of Trimethoprim for QAC-Adapted Subpopulations and Non-Adapted Control of L. monocytogenes Strains

Short range Minimum inhibitory concentration (MIC) assay was used to compare the susceptibilities against trimethoprim for QAC-adapted subpopulations and non-adapted control of *L. monocytogenes* as described previously [4] with minor modifications. QAC-adapted subpopulations and non-adapted control of *L. monocytogenes* strains were tested at trimethoprim concentration range of 0.125 to 1.5 µg/mL at 0.125 µg/mL intervals. Overnight grown cells were diluted in TSBYE to yield 10^7^ CFU/mL cell concentrations and 30 µL of diluted cell cultures were inoculated into the 96-well microtiter plate containing 270 µL of trimethoprim concentration yielding final cell inoculum of 10^6^ CFU/mL. The plates were incubated at 37 °C for 48 h and OD_600_ values were measured using microplate reader (ELx800 Absorbance Microplate Reader, BioTek, Winooski, VT, USA) after 48 h of incubation.

### 2.9. Determination of Growth Kinetics of QAC-Adapted Subpopulations and Non-Adapted Control of L. monocytogenes Strains in Trimethoprim-Containing Broth

The effect of QAC adaptation on the growth of *L. monocytogenes* strains in the presence of trimethoprim were monitored using growth kinetic assay. Briefly, an aliquot of 30 µL of overnight grown cells diluted to 10^7^ CFU/mL was inoculated in 96-well containing 270 µL of 0.125 μg/mL trimethoprim diluted in TSBYE to yield final cell inoculum of 10^6^ CFU/mL. The 96-well plate was incubated at 37 °C under shaker conditions (150 rpm) (C24 Classic series incubator shaker, New Brunswick Scientific Inc., Edison, NJ, USA) and bacterial growth was monitored every 4 h until 24 h at 600 nm (OD_600_) using microplate reader (ELx800 Absorbance Microplate Reader, BioTek, Winooski, VT, USA).

### 2.10. Determination of Survival of QAC-Adapted Subpopulations and Non-Adapted Control of L. monocytogenes Strains in Trimethoprim-Containing Agar

The heterologous tolerance response of QAC-adapted subpopulations and non-adapted control was compared by determining survivals on tryptic soy agar containing 0.6% yeast extract (TSAYE) containing 0.125 µg/mL trimethoprim. The QAC-adapted subpopulations and non-adapted control of *L. monocytogenes* strains were diluted to 10^7^ CFU/mL. Four consecutive decimal dilutions were performed and spotted on TSAYE containing 0.125 µg/mL trimethoprim and TSAYE without trimethoprim (control plate) and incubated at 37 °C for 48 h.

### 2.11. Statistical Analysis

Data were analyzed for statistical significance by a completely randomized design with 2 × 3 factorial structure (QAC-adapted and non-adapted stress vs. three methods of sublethal adaptation) in a randomized complete block design with replication considered as block. Significant differences in MIC of trimethoprim and the lag phase duration at OD_600_ between QAC-adapted subpopulations and non-adapted control were analyzed with the unpaired two-tailed t-test at three significance levels (*p* < 0.05; *p* < 0.01; and *p* < 0.001) using Microsoft Excel (Microsoft Excel, Version 2008). The fold increase and percentage increase in MIC of trimethoprim and fold increase in growth rate (OD_600_) were obtained using Duncan’s multiple range test (*p* < 0.05). Statistical analysis was performed using one-way ANOVA at significance level of *p* < 0.05. Logs transformed counts for survivals were analyzed using One-way ANOVA in a completely randomized block design and means were separated by Fisher’s protected LSD when *p* < 0.05. The statistical analysis was in this study was performed using the SAS software (SAS 9.4 TS Level 1M5; SAS Institute Inc., Cary, NC, USA).

## 3. Results

### 3.1. Changes in Short-Range MIC of Trimethoprim for QAC-Adapted Subpopulations of L. monocytogenes

The changes in MIC of trimethoprim for eight *L. monocytogenes* strains before and after sublethal exposure to daily cycles of fixed QAC concentration in broth (QAC-P1), or gradually increasing QAC concentration in broth (QAC-P2) and water model (QAC-P3) are shown in Figure 1. Before sublethal QAC exposure, *L. monocytogenes* strains exhibited trimethoprim MICs in the range of 0.25 to 0.67 µg/mL. After sublethal QAC exposure, the short-range MIC of trimethoprim for all strains of *L. monocytogenes* moved significantly higher to the range of 0.50 to 0.83 µg/mL for QAC-P1, 0.50 to 1.17 µg/mL for QAC-P2, and 0.42 to 1.0 µg/mL for QAC-P3. This increase in the short-range MIC of trimethoprim by 0.25 to 0.53 µg/mL of QAC-P1 (4/8 strains), or 0.34 to 0.67 µg/mL of QAC-P2 (4/8 strains), or 0.33 to 0.50 µg/mL of QAC-P3 (5/8 strains) was found to be significant (*p* < 0.05) as compared to non-adapted control. Such MIC increase of trimethoprim was equivalent to 1.7 to 2.5 fold (= 67–150%) higher for QAC-P1, or 1.8 to 2.3 fold (= 83–133%) higher for QAC-P2, or 1.7 to 2.0 fold (= 67–100%) higher for QAC-P3 after sublethal exposure to QAC which was significant compared to non-adapted control (*p* < 0.05) for these *L. monocytogenes* strains. For other strains, there was a 1.1 to 1.5 fold increase (= 11–50% higher) in short-range MIC of trimethoprim depending on the QAC-subpopulation.

This increase was equivalent to 1.7 to 2.5 fold (= 67–150%) higher in MIC for QAC-P1, or 1.8 to 2.3 fold (= 83–133%) higher in MIC for QAC-P2, or 2.0 fold (= 100%) higher in MIC for QAC-P3 as a result of sublethal exposure for 3/8 *L. monocytogenes* strains (NRRL B 33157, ATCC 43257 and ATCC 19116) which was significant compared to non-adapted control (*p* < 0.05) (Table 2).

There were also significant strain × subpopulation interactions in the short-range MIC of trimethoprim for some *L. monocytogenes* strains. For example, QAC-P1/QAC-P2 of strain EGD had 2.0 fold increase (= 100% higher) compared to QAC-P3 with 1.3 fold increase (= 25% higher) in short-range MIC of trimethoprim. ScottA had an increase of 2.0 fold (100%) higher in MIC of trimethoprim for QAC-P3 as compared to 1.3–1.4 fold (= 33–40%) higher for QAC-P1/QAC-P2. QAC-P1/QAC-P3 of N1-227 had 1.3–1.4 fold (= 33–40%) higher in MIC of trimethoprim compared to QAC-P2 with no change versus non-adapted control (Table 2).

### 3.2. Changes in Growth Rate of QAC-Adapted Subpopulations of L. monocytogenes in Trimethoprim-Containing Broth

The growth curves (OD_600_) of eight *L. monocytogenes* strains in trimethoprim before and after sublethal exposure to fixed or gradually increasing concentration of QAC are shown in Appendix A for QAC-P1, in Appendix A for QAC-P2 and in Appendix A for QAC-P3. There was a significant decrease in lag phase down to 8–12 h in trimethoprim-containing broth for 1/8 *L. monocytogenes* strains of QAC-P1 (N1-227, Appendix A) or QAC-P3 (EGD, Appendix A), compared to a longer lag phase of 12–22.7 h for non-adapted control (*p* < 0.05) (Figure 2). Meanwhile, for QAC-P2 of 3/8 *L. monocytogenes* strains (EGD, ScottA, and ATCC 19116) (Appendix A), there was a significant decrease in lag phase down to 4.7–7.3 h in trimethoprim compared to a longer lag phase of 10–16.7 h for non-adapted control (Figure 2). No differences in lag phase between QAC-adapted subpopulations were observed for other strains compared to non-adapted control.

A significant increase in growth rate (OD_600_) in trimethoprim was evident for 2/8 *L. monocytogenes* strains (EGD and ScottA) (Figure 3), which was 2.6 to 3.1 fold higher for QAC-P1, 3.9 to 4.8 fold higher for QAC-P2, or by 1.4 to 2.1 fold higher for QAC-P3 versus non-adapted control at 12 h (*p* < 0.05) (Table 2). Such increase in growth rate in trimethoprim of QAC-adapted subpopulations for these two strains continued to 20 h and 24 h time points (Table 2). With a few exceptions, no changes in growth rate in trimethoprim were observed for other *L. monocytogenes* strains compared to non-adapted control (Table 2).

There were some significant strain × subpopulation interactions for an increase in growth rate in trimethoprim. For example, at 10 h or 12 h time points, QAC-P2 of ScottA had 4.5 to 4.8 fold increase in growth rate compared to QAC-P1 or QAC-3 with 1.1 to 2.6 fold increase in trimethoprim (Table 2).

### 3.3. Changes in Survival of QAC-Adapted Subpopulations of L. monocytogenes in Trimethoprim-Containing Agar

The changes in survival (Log_10_ CFU/mL) determined by trimethoprim-containing agar for eight *L. monocytogenes* strains before and after sublethal exposure to fixed or gradually increasing concentration of QAC for QAC-P1, QAC-P2, and QAC-P3 are shown in Figure 4. Four patterns were observed for changes in survival: (1) QAC-adapted subpopulations of 3/8 *L. monocytogenes* strains (NRRL B 33155, NRRL B 33157, and ATCC 43527) exhibited highest increase in survival in trimethoprim which was by 1.9 to 2.8 log CFU/mL for QAC-P1 (Figure 4A), by 1.9 to 2.2 log CFU/mL for QAC-P2 (Figure 4B), or by 1.9 to 2.4 log CFU/mL for QAC-P3 (Figure 4C) (*p* < 0.05) compared to non-adapted control. This was equivalent to a percent increase in survival in trimethoprim by 28.7–41.6% for QAC-P1, or by 27.1–33.5% QAC-P2, or by 27.1–34.9% QAC-P3 (Table 2) for these 3/8 strains compared to non-adapted control. In this group, the subpopulations of NRRL B 33157 exhibited the highest survival increase; (2) QAC-adapted subpopulations of 2/8 *L. monocytogenes* strains (ScottA and ATCC 19117) had a significant 1.2 to 1.9 log CFU/mL (*p* < 0.05) increase in survival compared to non-adapted control. This was equivalent to a percent increase in survival by 17.5–28.2% compared to control; (3) QAC-adapted subpopulations of 1/8 *L. monocytogenes* strains (NRRL 33109) had no significant change in survival compared to non-adapted control; and (4) QAC-adapted subpopulations of 2/8 *L. monocytogenes* strains (EGD and N1-227) had a slight decrease in survival compared to non-adapted control.

When six strains belonging to serotype 4b were compared, QAC-adapted subpopulations for four of those strains (NRRL B 33157, NRRL B 33155, ATCC 43527, and ScottA) exhibited the highest percent increase in survival, followed by that of serotype 4c (ATCC 19116). QAC-adapted subpopulations of one 4b strain (NRRL B 33109) had no change in survival, while a reduction in survival was observed for another 4b strain (N1-227) compared to the non-adapted control.

There were some significant strain × subpopulation interactions for an increase in survival for some *L. monocytogenes* strains. For example, QAC-P2 of EGD had the highest percent survival increase (20%) compared to QAC-P1 or QAC-P3 in trimethoprim.

When the trimethoprim level increased from 0.125 µg/mL to 0.25 and 0.5 µg/mL in agar, all QAC-adapted subpopulations were non-detectable for all eight strains of *L. monocytogenes* similar to that of non-adapted control (data not shown).

## 4. Discussion

Antibiotic resistance is slowly emerging worldwide in foodborne isolates of *L. monocytogenes*. Although the antibiotic resistant *L. monocytogenes* remains low, there has been a considerable increase in isolates demonstrating resistance to one or more antibiotics. The first multidrug resistant strain of *L. monocytogenes* was reported in France in 1988 and since then isolates from various food, clinical and environmental samples resistant to single or multiple antibiotics have been described [15]. For example, *L. monocytogenes* isolates obtained from a commercial poultry processing plant in Georgia, USA, were found to be tolerant to seven antibiotics including, ceftriaxone, ciprofloxacin, oxacillin, tetracycline, clindamycin, linezolid and trimethoprim/sulfamethoxazole [19]. In Malaysia, resistance to penicillin, meropenem, and rifampicin was demonstrated in *L. monocytogenes* isolates from vegetable farms and retail markets [20]. In Morocco, resistance to cefotaxime, sulfonamide, nalidixic acid, Fosfomycin, and lincosamide was detected in *L. monocytogenes* from food samples [21]. In Poland, *L. monocytogenes* strains isolated from fish processing plants were found to be resistant to erythromycin and trimethoprim/sulfamethoxazole [22]. In Brazil, *L. monocytogenes* strains from a pig slaughterhouse were resistant against ampicillin, penicillin, and trimethoprim/ sulfamethoxazole [23].

In food processing environments, disinfectants are routinely used at 50–100 times higher concentrations than that of the minimum bactericidal concentration (MBC) against foodborne bacterial pathogens. However, the bacteria trapped in the environmental niches like cracks or on inside equipment are exposed to a sublethal biocide concentration due to dilution caused by the presence of water or other organic matter. Residues of the disinfectant may remain on the surface due to insufficient rinsing after cleaning step [4]. Recent studies suggest that the exposure to sublethal biocide concentrations may select for resistance against biocides as well as cross-resistance to antibiotics [24].

Currently, there is a slow emergence of resistance to trimethoprim in *L. monocytogenes*. After the first discovery of *L. monocytogenes* isolate exhibiting a high-level resistance against trimethoprim in a clinical sample [16], more strains with trimethoprim tolerance have been later isolated from food processing environments [17]. A total of 33 *L. monocytogenes* isolated from ready to eat meat products from the deli section of 12 supermarkets and 12 open-air markets in Nanjing, China, exhibited resistant against trimethoprim/sulfamethoxazole (≥4 µg/mL) [25]. Currently, the overall incidence of trimethoprim resistance in *L. monocytogenes* remains very low. *L. monocytogenes* strains are naturally susceptible to trimethoprim and acquired antimicrobial resistance in clinical strains is rare [16,17,26,27]. However, by using three approaches for creating a continuous exposure to sublethal concentrations of QAC against actively growing planktonic cells of *L. monocytogenes*, and subsequently evaluating for antibiotic susceptibility against trimethoprim, we detected significant increase in: (1) the short-range MIC of trimethoprim in QAC-adapted subpopulations; (2) the growth rate of QAC-adapted subpopulations in trimethoprim-containing broth model; and (3) the survival of QAC-adapted subpopulations in trimethoprim-containing agar model. Our results show that there is a potential for the development of low-level tolerance to trimethoprim in *L. monocytogenes* strains after continuous sublethal exposure to QAC. A significant increase in short range MIC of trimethoprim for QAC-adapted subpopulations was observed by 1.7 to 2.5 fold (from the initial 0.5 to final 1 µg/mL) for *L. monocytogenes*. A similar two fold increase in MIC of trimethoprim was also observed for one of the clinical *L. monocytogenes* isolates SLCC2540 (serotype 3b) adapted to sequentially increasing concentration of QAC in other recent studies in Germany [28]. After QAC adaptation, the MIC of trimethoprim for the clinical *L. monocytogenes* isolate SLCC2540 increased from 0.0312 to 0.0625 µg/mL. In other species such as *Pseudomonas aeruginosa* when exposed to triclosan, it selected for mutants exhibiting greater than 32 fold increase in MIC of trimethoprim from 32 to 1024 µg/mL [29].

While the low-level increase in MIC of trimethoprim against QAC-adapted subpopulations observed in this study was statistically significant, it was below the CLSI break point for trimethoprim against *L. monocytogenes* [30]. However, such small increases that are not considered clinically relevant must not be disregarded since subpopulations exhibiting low-level tolerance are likely to survive antibiotic exposure than the non-adapted cell population [31]. Generally, MIC is used as a gold-standard measure of the bacterial susceptibility against the respective drug. However, the relationships between the antimicrobial concentration and its pharmacodynamic (PD) effect on the bacterial population is complicated. Such dynamic changes cannot be determined using MIC as a parameter alone [32]. Therefore, we employed growth kinetic studies to determine the effect of sublethal trimethoprim on adapted cells of *L. monocytogenes*. QAC-adapted subpopulations exhibited a significantly shorter lag phase in trimethoprim as compared to the non-adapted strains. This was expressed as a decrease in lag phase down to 4 h in sublethal trimethoprim for QAC-adapted subpopulations as compared to 8 h for the non-adapted control. In this study, we did not test the effect of higher concentrations of trimethoprim on the extension of lag phase and its influence on the QAC-adapted *L. monocytogenes* strains. To develop tolerance, any changes in the lag time was the first significant change made by bacteria in response to antibiotic [33]. Therefore, the quantification of the lag phase duration facilitates the understanding of bacterial interactions with antibiotics and predicts the bacterial response at different antibiotic concentrations. Since the extension of lag time is dose dependent, such a lag phase extension provides effective protection for the bacteria [34].

In order to determine the bacterial subpopulation that may be demonstrating the potential evolution of tolerance development, we isolated the subpopulations of adapted cells of *L. monocytogenes* by plating on trimethoprim-containing agar model. A significant increase in survival by 1–3 log CFU/mL was observed for QAC-adapted subpopulations of *L. monocytogenes* in the presence of trimethoprim-containing agar. The surviving CFU of QAC-adapted subpopulations of *L. monocytogenes* isolated on trimethoprim-containing agar were preserved for further work by whole genome analysis or heteroresistance. Heteroresistance is described as the presence of subpopulations of bacterial cells within an isogenic population that demonstrates higher levels of antibiotic resistance. The population analysis profiling (PAP) method is used to determine heteroresistance in a bacterial population subjected to antibiotic increments using spread plate techniques for CFU counting. Previously, PAPs were performed to determine heteroresistance in methicillin-resistant *Staphylococcus aureus* (MRSA) isolates using brain heart infusion (BHI) plates containing various amounts of ceftaroline. Out of the 57 isolates tested for *S. aureus*, PAPS disclosed heteroresistance among 12 isolates. Originally, the isolates of *S. aureus* had an MIC of 1 μg/mL by broth dilution assay, however, PAPs determined the growth at ceftaroline concentrations of 1.25 to 3 μg/mL. The frequencies of such resistant subpopulations were 1 subclone in 10^4^ to 10^5^ colonies [35]. Overall, the therapeutic dosing can select for such resistant subpopulation from the majority. This is particularly true in cases where a small subset of population is resistant in a generic antibiotic-sensitive population, which remains undetectable through the criteria set for traditional *in vitro* antibiotic susceptibility testing [36]. Therefore, biocide induced subpopulations isolated on trimethoprim-containing agar in our studies are useful for exploring the different mechanisms behind the low-level tolerance to trimethoprim in *L. monocytogenes* strains. Also, the stability of the QAC-induced trimethoprim tolerance in the subpopulations of *L. monocytogenes* needs to be determined.

In summary, we explored the link between QAC adaptation and development of low-level tolerance to trimethoprim in some strains of *L. monocytogenes* isolated from food sources. Although we did not address the mechanisms underlying adaptation to QAC and trimethoprim tolerance development, this work will provide valuable guidance to expand to different environmental conditions where QAC effluents may persist in the food processing environments for a broader understanding of this phenomenon. Besides trimethoprim, we observed the development of low-level tolerance to fluoroquinolone antibiotic ciprofloxacin in QAC-adapted subpopulations of *L. monocytogenes* strains which was published recently [37]. The other recent studies indicate that such low-level tolerance to antibiotics can accelerate the emergence of resistance [38], therefore, the data from this study is a first step in deducing any changes against trimethoprim resistance in *L. monocytogenes*. Based on a limited number of strains tested, we did not observe the development of clinically relevant antibiotic resistance against trimethoprim in QAC-adapted subpopulations of *L. monocytogenes*. Therefore, further work is needed to determine the effect of biocide-induced sublethal stresses on foodborne bacterial pathogens and subsequent cross-resistance to antibiotics using a wide variety of naturally occurring isolates of *L. monocytogenes* to address the risk posed by the QAC residues and effluents from food processing environments on antibiotic tolerance response.

In conclusion, our work presented evidence for low-level tolerance to one of the alternate antibiotics used for listeriosis treatment in some *L. monocytogenes* strains when exposed to sublethal doses of QAC, a widely used biocide in the food industries and hospitals. Our data shows that some *L. monocytogenes* strains adapted to QAC may have a survival advantage in certain antibiotics during the early growth stages which is valuable for pathogen risk assessment. Besides employing broth models using defined growth media, our study also used water samples that were spiked with varying known concentrations of QAC for better understanding of *L. monocytogenes* adaptation. As part of our future work, we will test water samples after sanitation for residual QAC levels and the presence of naturally surviving QAC-adapted and low-level antibiotic-tolerant *L. monocytogenes* strains persisting in food residues. We will also explore the correlation between both homologous and heterologous tolerance in *L. monocytogenes* after QAC adaptation with respect to selected antibiotics. Antibiotic resistance in foodborne isolates of *L. monocytogenes* is slowly increasing in many countries. While we did not find the clinical level of resistance to trimethoprim to be high enough to compromise the treatment of human listeriosis, a low-level tolerance to trimethoprim in QAC-adapted subpopulations may raise the possibility of a future acquisition of resistance by *L. monocytogenes*. Therefore, our work may lead to early detection of low-level antibiotic tolerance in food isolates of *L. monocytogenes* induced by QAC-adaptation based on early growth dynamics. This information is useful for improving the sanitation steps and to eliminate safety concerns by preventing and mitigating the formation of low-level antibiotic tolerant strains if persisting in some food production and food processing environments.

## Figures and Tables

**Figure 1 foods-10-01800-f001:**
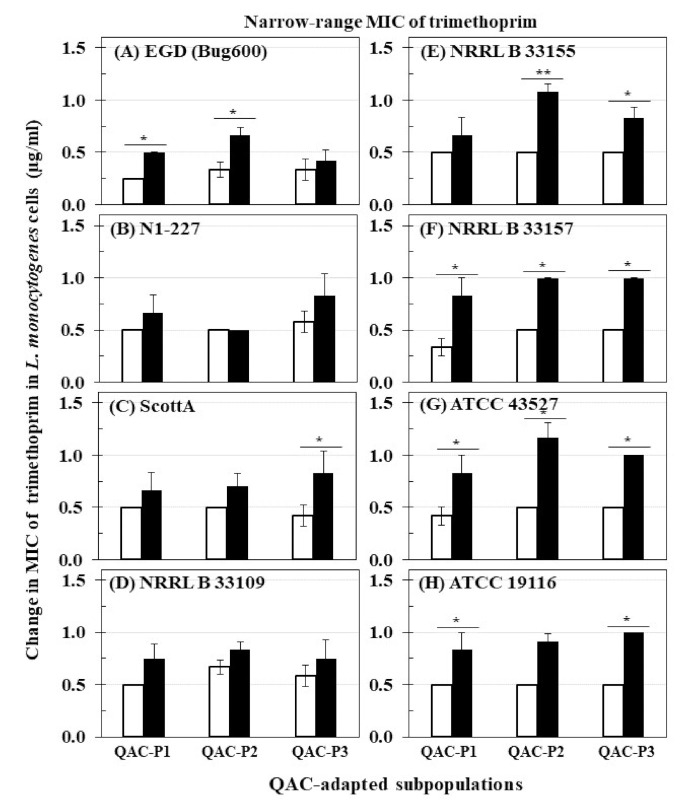
Changes in minimum inhibitory concentration (MIC) of trimethoprim (µg/mL) for QAC-adapted subpopulations (closed bars) compared to non-adapted controls (NAc) (open bars) of eight *L. monocytogenes* strains: (**A**) EGD (Bug600), (**B**) N1-227, (**C**) ScottA, (**D**) NRRL B 33109, (**E**) NRRL B 33155, (**F**) NRRL B 33157, (**G**) ATCC 43527, (**H**) ATCC 19116. Error bars indicate standard errors of means. Statistically significant *p* values are indicated by asterisks (*, *p* < 0.05; **, *p* < 0.01) that were obtained using the unpaired two-tailed *t*-test.

**Figure 2 foods-10-01800-f002:**
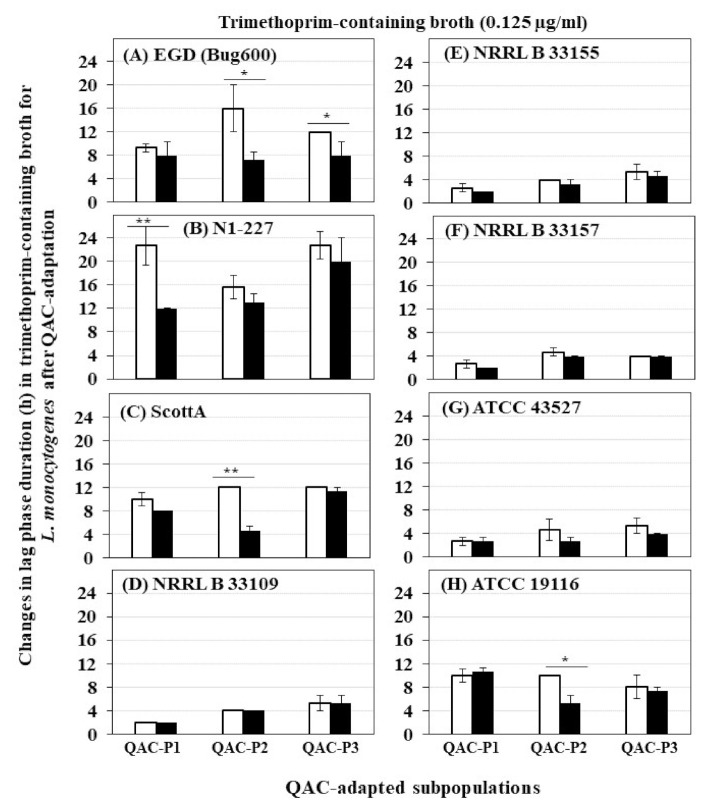
Changes in lag phase duration (h) for three QAC-adapted subpopulations (closed bar) in trimethoprim-containing broth (0.125 µg/mL) compared to non-adapted controls (NAc) (open bars) of eight *L. monocytogenes* strains at 37 °C: (**A**) EGD (Bug600), (**B**) N1-227, (**C**) ScottA, (**D**) NRRL B 33109, (**E**) NRRL B 33155, (**F**) NRRL B 33157, (**G**) ATCC 43527, (**H**) ATCC 19116. Error bars indicate standard errors of means. Statistically significant *P* values are indicated by asterisks (*, *p* < 0.05; **, *p* < 0.01) that were obtained using the unpaired two-tailed *t*-test.

**Figure 3 foods-10-01800-f003:**
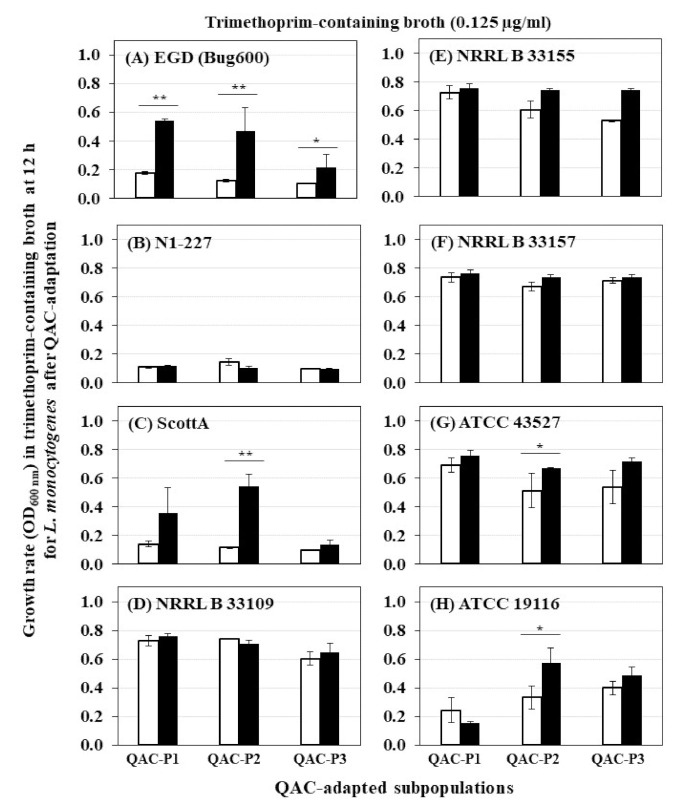
Changes in growth rate at 12 h (OD_600_) in trimethoprim-containing broth of QAC-adapted subpopulations (closed bars) compared to non-adapted control (open bars) cells of eight *L. monocytogenes* at 37 °C: (**A**) EGD (Bug600), (**B**) N1-227, (**C**) ScottA, (**D**) NRRL B 33109, (**E**) NRRL B 33155, (**F**) NRRL B 33157, (**G**) ATCC 43527, (**H**) ATCC 19116. Error bars indicate standard errors of means. Statistically significant *P* values are indicated by asterisks (*, *p* < 0.05; **, *p* < 0.01) that were obtained using the unpaired two-tailed *t*-test.

**Figure 4 foods-10-01800-f004:**
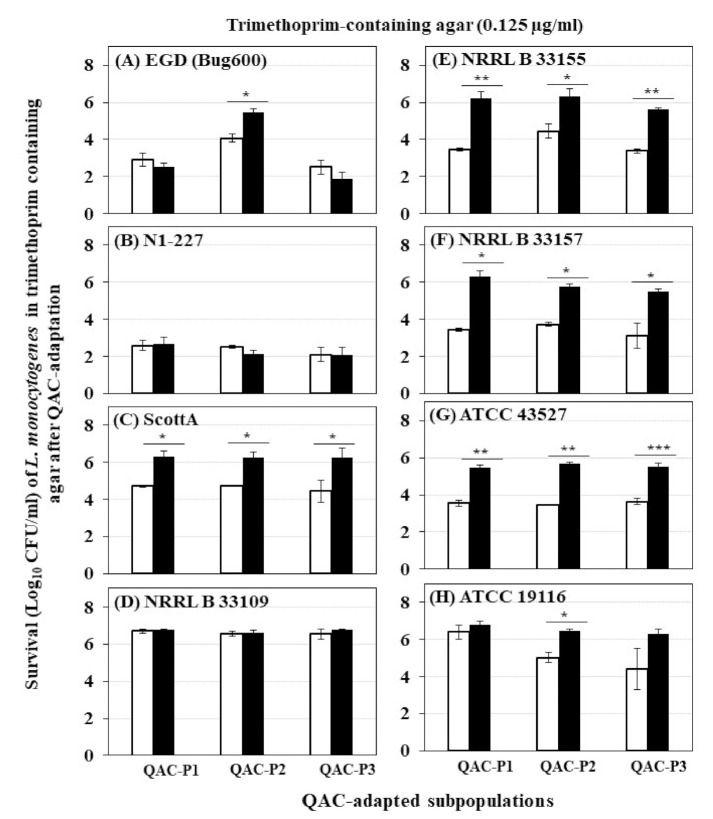
Survival (Log_10_ CFU/mL) of three QAC-adapted subpopulations (closed bars) and non-adapted control (open bars) of eight *L. monocytogenes* strains in trimethoprim-containing agar: (**A**) EGD (Bug600), (**B**) N1-227, (**C**) ScottA, (**D**) NRRL B 33109, (**E**) NRRL B 33155, (**F**) NRRL B 33157, (**G**) ATCC 43527, (**H**) ATCC 19116. Error bars indicate standard errors of means. Statistically significant *P* values are indicated by asterisks (*, *p* < 0.05; **, *p* < 0.01; ***, *p* < 0.001) that were obtained using the unpaired two-tailed *t*-test.

**Table 1 foods-10-01800-t001:** *L. monocytogenes* strains used in this study.

Designation	Lineage	Serotype	Source	Isolation Source
N1-227	I	4b	CDC, Atlanta	Food epidemic (food isolate from hotdog associated with US outbreak in 1998–1999)
ATCC 19116	III	4c	University of Wisconsin	Poultry
ScottA	I	4b	FDA	Human epidemic (clinical isolate associated with Massachusetts listeriosis outbreak in 1983)
EGD (Bug600)	II	1/2a	Institute Pasteur	Guinea pigs
NRRL B-33109	I	4b	USDA-ARS, NADC	Cooler condenser
NRRL B-33155	I	4b	USDA-ARS, NADC	Sodium caseinate epidemic strain, CA, 1985 outbreak
NRRL B-33157	I	4b	USDA-ARS, NADC	Insect debris found in cheese plant
ATCC 43257	I	4b	CDC, Atlanta	Mexican Style cheese, CA

**Table 2 foods-10-01800-t002:** Fold increase in MIC of trimethoprim, fold increase in growth rate at different time points (OD_600_) in trimethoprim-containing broth, and percent increase in survival in trimethoprim-containing agar of three QAC-adapted subpopulations (QAC-P1, QAC-P2, and QAC-P3 compared to non-adapted control (NAc) of eight *L. monocytogenes* strains.

Listeria monocytogenes Strains	Fold Increase ^1^ in MIC of Trimethoprim for QAC-P1 ± SE	Fold Increase ^2^ in Growth (OD_600_) of QAC-P1 Compared to Non-Adapted Control in Trimethoprim-Containing Broth at Different Time Points				Percentage Increase ^3^ in Survivals of Trimethoprim-Agar for QAC-P1 ± SE
		10 h	12 h	20 h	24 h	
EGD (Bug600)	2.0 ± 0.0 ^a^	1.7 ± 0.5 ^a^	3.1 ± 0.2 ^a^	1.4 ± 0.4 ^a^	1.2 ± 0.2 ^b^	−6.1 ± 3.1 ^d^
N1 227	1.3 ± 0.3 ^b^	1.1 ± 0.1 ^a^	1.1 ± 0.1 ^b^	1.7 ± 0.2 ^a^	2.0 ± 0.2 ^a^	1.2 ± 1.1 ^c^
Scott A	1.3 ± 0.3 ^b^	1.4 ± 0.1 ^a^	2.6 ± 0.9 ^a^	1.0 ± 0.1 ^b^	1.2 ± 0.1 ^b^	23.4 ± 4.5 ^b^
NRRL B 33109	1.5 ± 0.3 ^b^	1.3 ± 0.4 ^a^	1.0 ± 0.1 ^b^	1.1 ± 0.1 ^b^	1.0 ± 0.0 ^b^	0.6 ± 2.1 ^c^
NRRL B 33155	1.3 ± 0.3 ^b^	1.1 ± 0.1 ^a^	1.0 ± 0.1 ^b^	1.0 ± 0.1 ^b^	1.0 ± 0.0 ^b^	40.9 ± 4.3 ^a^
NRRL B 33157	2.5 ± 0.7 ^a^	1.1 ± 0.1 ^a^	1.0 ± 0.1 ^b^	1.0 ± 0.1 ^b^	1.0 ± 0.0 ^b^	41.1 ± 4.5 ^b^
ATCC 43257	2.0 ± 0.0 ^a^	1.1 ± 0.2 ^a^	1.1 ± 0.1 ^b^	1.0 ± 0.1 ^b^	1.0 ± 0.0 ^b^	28.7 ± 3.0 ^b^
ATCC 19116	1.7 ± 0.3 ^a^	0.5 ± 0.3 ^b^	0.6 ± 0.2 ^b^	1.0 ± 0.2 ^b^	1.2 ± 0.2 ^b^	16.9 ± 8.7 ^b^
**Listeria monocytogenes Strains**	**Fold Increase ^1^ in MIC of Trimethoprim for QAC-P2 ± SE**	**Fold Increase ^2^ in Growth (OD_600_) of QAC-P2 Compared to Non-Adapted Control in Trimethoprim-Containing Broth at Different Time Points**				**Percentage Increase ^3^ in Survivals of** **Trimethoprim-Agar for QAC-P2 ± SE**
		**10 h**	**12 h**	**20 h**	**24 h**	
EGD (Bug600)	2.0 ± 0.4 ^a^	2.1 ± 0.5 ^b^	3.9 ± 1.1 ^a^	4.3 ± 0.2 ^a^	1.9 ± 1.0 ^a^	20.7 ± 6.1 ^a^
N1 227	1.0 ± 0.0 ^b^	0.9 ± 0.0 ^c^	0.7 ± 0.1 ^b^	1.8 ± 0.2 ^c^	1.7 ± 0.4 ^a^	−5.8 ± 3.6 ^c^
Scott A	1.4 ± 0.3 ^a^	4.5 ± 0.1 ^a^	4.8 ± 0.4 ^a^	2.5 ± 0.0 ^b^	1.8 ± 0.2 ^a^	21.9 ± 4.4 ^a^
NRRL B 33109	1.3 ± 0.3 ^b^	0.9 ± 0.4 ^c^	0.9 ± 0.0 ^b^	0.9 ± 0.0 ^d^	0.9 ± 0.0 ^b^	0.70 ± 0.4 ^b^
NRRL B 33155	2.2 ± 0.2 ^a^	1.2 ± 0.0 ^b^	1.2 ± 0.1 ^b^	1.0 ± 1.0 ^d^	1.0 ± 0.0 ^b^	27.1 ± 1.4 ^a^
NRRL B 33157	2.0 ± 0.0 ^a^	1.2 ± 0.1 ^b^	1.1 ± 0.0 ^b^	0.9 ± 0.0 ^d^	1.0 ± 0.0 ^b^	30.9 ± 1.3 ^a^
ATCC 43257	2.3 ± 0.3 ^a^	1.3 ± 0.4 ^b^	1.3 ± 0.4 ^b^	1.2 ± 0.1 ^d^	1.0 ± 0.1 ^b^	33.5 ± 1.2 ^a^
ATCC 19116	1.8 ± 0.2 ^a^	3.5 ± 0.8 ^a^	1.7 ± 0.7 ^b^	1.1 ± 0.1 ^d^	1.0 ± 0.0 ^b^	22.2 ± 5.2 ^a^
**Listeria monocytogenes Strains**	**Fold Increase ^1^ in MIC of Trimethoprim for QAC-P3 ± SE**	**Fold Increase ^2^ in Growth (OD_600_) of QAC-P3 Compared to Non-Adapted Control in Trimethoprim-Containing Broth at Different Time Points**				**Percentage Increase ^3^ in Survivals of** **Trimethoprim-Agar for QAC-P3 ± SE**
		**10 h**	**12 h**	**20 h**	**24 h**	
EGD (Bug600)	1.3 ± 0.3 ^b^	1.5 ± 0.4 ^a^	2.1 ± 0.8 ^a^	2.1 ± 0.4 ^a^	2.3 ± 0.4 ^a^	−9.6 ± 5.5 ^c^
N1 227	1.4 ± 0.4 ^a^	1.0 ± 0.0 ^a^	1.0 ± 0.0 ^b^	1.2 ± 0.3 ^a^	1.2 ± 0.1 ^b^	−0.2 ± 0.1 ^b^
Scott A	2.0 ± 0.0 ^a^	1.1 ± 0.0 ^a^	1.4 ± 0.3 ^a^	1.9 ± 0.9 ^a^	1.2 ± 0.2 ^b^	27.1 ± 6.7 ^a^
NRRL B 33109	1.3 ± 0.1 ^b^	1.0 ± 1.5 ^a^	1.2 ± 1.7 ^b^	1.0 ± 0.1 ^b^	0.9 ± 0.1 ^b^	2.5 ± 2.3 ^b^
NRRL B 33155	1.7 ± 0.2 ^a^	1.2 ± 1.8 ^a^	1.4 ± 1.6 ^b^	1.1 ± 0.1 ^a^	1.0 ± 0.0 ^b^	32.6 ± 0.9 ^a^
NRRL B 33157	2.0 ± 0.0 ^a^	1.0 ± 0.1 ^a^	1.0 ± 0.0 ^b^	1.0 ± 0.0 ^b^	0.9 ± 0.0 ^b^	35.0 ± 6.1 ^a^
ATCC 43257	2.0 ± 0.0 ^a^	1.6 ± 0.9 ^a^	1.3 ± 0.4 ^b^	1.1 ± 0.1 ^a^	1.1 ± 0.1 ^b^	28.1 ± 2.7 ^a^
ATCC 19116	2.0 ± 0.0 ^a^	1.2 ± 0.4 ^a^	1.2 ± 1.2 ^b^	1.2 ± 0.1 ^a^	1.1 ± 0.1 ^b^	28.2 ± 8.2 ^a^

^a–d^ Different letters indicate significant differences between means (*p* < 0.05) by Duncan’s multiple range test for ^1^ Fold Increase in MIC of Trimethoprim for QAC-Adapted Subpopulations, or ^2^ Fold Increase in Growth (OD_600 nm_) of QAC-Adapted Subpopulations Compared to Non-Adapted Control in Trimethoprim-Containing Broth at Different Time Points, or ^3^ Percentage Increase in Survivals in Trimethoprim-Containing Agar for QAC-Adapted Subpopulations.

## Data Availability

All relevant data are within the paper.

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
