# Peer review of "Low-Level Tolerance to Antibiotic Trimethoprim in QAC-Adapted Subpopulations of Listeria monocytogenes"

_foods, 2021, doi:10.3390/foods10081800_

Round 1
Reviewer 1 Report
The manuscript entitled: “Low-level tolerance to antibiotic trimethoprim in QAC- adapted subpopulations of Listeria monocytogenes” presents an interesting study regarding the investigation of tolerance development to trimethoprim in Listeria monocytogenes strains. The results of this work are very useful in understanding the predisposition to appearance trimethoprim-resistant strains of L. monocytogenes in different environments that have a wide implication in the clinic and food processing industry.
- Define all the abbreviations at the first use in the abstract also and then in the text, eg. MIC
- Introduction is too long, I suggest shortening and move some information in the discussion section.
- Define DMSO – subchapter 2.7; OD600 – subchapter 2.4;
- In table 2 legend you should clearly specify what means a, b, c letters. You specify only in the note that different letters indicate significant differences but not what exactly means a, b, and c
- In Figure 2 and 3 legends, you define *** P<0.001, but this can not be seen in the figure. Can you check if something is missing?
- In the discussions section, it has to clearly emphasize the strengths and limitations of this study.
Reviewer 1 Report
Authors Responses
The manuscript entitled: “Low-level tolerance to antibiotic trimethoprim in QAC- adapted subpopulations of Listeria monocytogenes” presents an interesting study regarding the investigation of tolerance development to trimethoprim in Listeria monocytogenes strains. The results of this work are very useful in understanding the predisposition to appearance trimethoprim-resistant strains of L. monocytogenes in different environments that have a wide implication in the clinic and food processing industry.
1. Define all the abbreviations at the first use in the abstract also and then in the text, eg. MIC
Author’s response: All abbreviations have been defined at the first use in the abstract and in other sections of the revised manuscript.
2. Introduction is too long, I suggest shortening and move some information in the discussion section.
Author’s response: Based on the reviewer’s suggestion, the introduction section has been shortened by moving a part of the information into the discussion section in the revised manuscript.
3. Define DMSO – subchapter 2.7; OD600 – subchapter 2.4;
Author’s response: These two abbreviations are defined in the revised manuscript.
4. In table 2 legend you should clearly specify what means a, b, c letters. You specify only in the note that different letters indicate significant differences but not what exactly means a, b, and c
Author’s response: Table 2 legend has been updated in the revised manuscript.
5. In Figure 2 and 3 legends, you define *** P<0.001, but this cannot be seen in the figure. Can you check if something is missing?
Author’s response: P<0.001 has been deleted in the Figures 2 and 3 legends of the revised manuscript.
6. In the discussions section, it has to clearly emphasize the strengths and limitations of this study.
Author’s response: Based on the reviewer’s suggestion, we have now clearly emphasized the key strengths and limitations of this study towards the end of the discussion section of the revised manuscript.
We are thankful to the reviewer for the excellent suggestions on this manuscript. We are grateful for the critical reading and review.
Ramakrishna Nannapaneni, Corresponding Author
Submission Date: 10 July 2021

Reviewer 2 Report
The manuscript entitled: Low-level tolerance to antibiotic trimethoprim in QAC-adapted subpopulations of Listeria monocytogenes” reports data on antibiotic resistance to microorganism from food samples. The manuscript title should refer to this aspect as a proirity considering that the manuscript is sent to “Foods”. The Introduction section is well asessed and clear properly the terms of the problem. Nonetheless, the experimental part reports data on Listeria monocytogenes strains and growth conditions from different sources (eight in total). More detailed information on foodstuff sources should be added to clear te context and the regulation existing should be mentioned and information given. The risk aspects connected to this microorganism should be commented also and references given. The food processing used disinfectants amounts should be given (refer to existing regulations) as well as the clean samples data. As it is, lacking a “to the ppoint” Conlusion section, the manuscript seems more appropriate to a microbiology Journal.
It should be carefully decribed the term of the problem in the food area and the novelty of the findings together with any possible impact on the area on interest. The statistical analysis would require more data for better assessment. The manuscript is in general also difficult to read and should be simplified where possible.
Reviewer 2 Report
Authors Responses
The manuscript entitled: Low-level tolerance to antibiotic trimethoprim in QAC-adapted subpopulations of Listeria monocytogenes” reports data on antibiotic resistance to microorganism from food samples. The manuscript title should refer to this aspect as a priority considering that the manuscript is sent to “Foods”. The Introduction section is well assessed and clear properly the terms of the problem. Nonetheless, the experimental part reports data on Listeria monocytogenes strains and growth conditions from different sources (eight in total). More detailed information on foodstuff sources should be added to clear the context and the regulation existing should be mentioned and information given. The risk aspects connected to this microorganism should be commented also and references given.
Author’s response: Based on the reviewer’s suggestion, we have added more information regarding the food sources and on regulations concerning L. monocytogenes in the introduction section of the revised manuscript.
The food processing used disinfectants amounts should be given (refer to existing regulations) as well as the clean samples data.
Author’s response: Based on the reviewer’s suggestion, we have included more information on the disinfectant concentrations routinely used in the food industries in the introduction section of the revised manuscript.
As it is, lacking a “to the point” Conclusion section, the manuscript seems more appropriate to a microbiology Journal.
Author’s response: Based on the reviewer’s suggestion, we have summarized, to the point, one main conclusion of this manuscript in a single sentence at the beginning of the last paragraph of the revised discussion section.
It should be carefully described the term of the problem in the food area and the novelty of the findings together with any possible impact on the area on interest. The statistical analysis would require more data for better assessment.
Author’s response: Based on the reviewer’s suggestion, we have carefully described the nature of this problem in the food area in the beginning of both introduction and discussion sections, and now clearly summarized the strengths, limitations, further experiments, impact, and novelty of this work in the last paragraph of the discussion section in the revised manuscript.
The manuscript is in general also difficult to read and should be simplified where possible.
Author’s response: Based on the reviewer’s suggestion, we have carefully reorganized the flow of the key ideas that are easy to understand in the revised manuscript.
We are thankful to the reviewer for the excellent suggestions on this manuscript. We are grateful for the critical reading and review.
Ramakrishna Nannapaneni, Corresponding Author
Submission Date: 10 July 2021

Round 2
Reviewer 1 Report
The authors addressed all my comments. I endorse the publication.
Author Response
Reviewer 1 (Round 2) Report
Authors Responses
The authors addressed all my comments. I endorse the publication.
Author’s response: We are thankful to the reviewer for the excellent suggestions on this manuscript. We are grateful for the critical reading and review.
Ramakrishna Nannapaneni, Corresponding Author
Submission Date: 18 July 2021

Reviewer 2 Report
The manuscript has been modified, nonetheless the end points seems still concerning the microbiological aspects while the impact on foodstuff seems still not properly evidenced. Any food sample has been tested? Please assess these aspects and ptential impact on the area. Please exploit better the scope and impact on food and food derivatives on the proposed manuscript and the contribution to the field. Moreover, please evidence also in the text all the modifications made for an easy to read check. In Table 1 please specify the term "food epidemic". A Conclusion section should be useful to assess the end points of the manuscript.
Author Response
Reviewer 2 (Round 2) Report
Authors Responses
The manuscript has been modified, nonetheless the end points seems still concerning the microbiological aspects while the impact on foodstuff seems still not properly evidenced. Any food sample has been tested? Please assess these aspects and potential impact on the area. Please exploit better the scope and impact on food and food derivatives on the proposed manuscript and the contribution to the field. Moreover, please evidence also in the text all the modifications made for an easy to read check. In Table 1 please specify the term "food epidemic". A Conclusion section should be useful to assess the end points of the manuscript.
Author’s response: Based on the reviewer’s suggestion, we have added a new conclusion paragraph to explain the broad findings, the nature of samples tested, and the full scope and impact of this work. Please see the track changes file that shows the changes made in Table 1 on page 3 and the new conclusion paragraph with contribution to the field on page 14.
We are thankful to the reviewer for the excellent suggestions on this manuscript. We are grateful for the critical reading and review.
Ramakrishna Nannapaneni, Corresponding Author
Submission Date: 18 July 2021

Round 3
Reviewer 2 Report
The manuscript has been modified, nonetheless the clear reference to food is still to be cleared. Please comment this point since at first sight the manuscript seems still mainly focused on microbiology starting from the Abstract. The esd in table is reported in some cases as 0.0, is it correct? The Conclusion section should be a separate paragraph with perspective poit of view of the Authors and limits of the proposed work putting in evidence the end points and application of the findings in the food area.
Author Response
Reviewer 2 (Round 3) Report
Authors Responses
The manuscript has been modified, nonetheless the clear reference to food is still to be cleared. Please comment this point since at first sight the manuscript seems still mainly focused on microbiology starting from the Abstract.
Author’s response: Based on the reviewer’s suggestion, we have added new statements in the abstract and in the introduction sections of the revised manuscript to bring up a clear refence to the food and the novelty of our work. See explained below.
Below are the new statements added in the beginning of the abstract on page 1:
Between January to July 2021, there were as many as 30 recalls in the USA due to potential Listeria monocytogenes contamination from a variety of food products including muffins, kimchi, chicken salad, ready-to-eat chicken, smoked fish, mushrooms, queso fresco cheese, ice cream, turkey sandwiches, squash, and other foods. Such contaminated food chain can serve as a potential vehicle for transmitting antibiotic resistant bacteria since there is a slow emergence of multi-drug antibiotic resistance in L. monocytogenes. Biocides are essential for safe food processing, but they may also induce unintended selective pressure at sublethal doses for the expression of antibiotic resistance in L. monocytogenes. To better understand the sources of such slow emergence of antibiotic resistance through biocide residues present in the food environments, we are working on the role of sublethal doses of commonly used biocides in defined broth and water models for understanding L. monocytogenes adaptation.
Below are the new statements added towards the end of the abstract on page 1:
These experimental models are useful in developing early detection methods for tracking the slow emergence of antibiotic tolerant strains through food chain. Also, these findings are useful in understanding the predisposing conditions leading to slow emergence of antibiotic resistant strains of L. monocytogenes in various food production and food processing environments.
Below are the new statements added in the introduction section on page 2 in round 2:
Listeria monocytogenes is an intracellular foodborne pathogen of high-risk public health concern. This pathogen has been isolated from various ready-to-eat food products and from food-contact processing environment such as chillers, conveyors, slicers, pack-aging machine, and freezers. To combat this pathogen, US regulatory agencies established a ‘zero-tolerance policy’ against L. monocytogenes in ready-to- eat foods. Recently in July 2021, 8.5 million pounds of ready-to-eat chicken products (frozen, fully cooked) were re-called by USDA Food Safety and Inspection Service because of contamination with L. monocytogenes after three illnesses and one death was reported in the USA (CDC, 2021a). Also, another recent listeriosis outbreak in the USA in May 2021 was associated with the consumption of queso fresco cheese where there were twelve hospitalization and one death (CDC, 2021b).
Below are the new statements added in the above paragraph in the introduction section on page 2 in round 3:
In addition to these recent outbreaks, between January to July 2021 FDA has reported 30 food safety recalls in the USA due to potential L. monocytogenes contamination in a variety of food products including muffins, kimchi, chicken salad, ready-to-eat chicken, smoked fish, mushrooms, queso fresco cheese, ice cream, turkey sandwiches, squash, and other foods. Besides USA, European data also show that there is in increase in the number of confirmed listeriosis cases in the European Union (EFSA and ECDC, 2019).
The esd in table is reported in some cases as 0.0, is it correct?
Author’s response: Yes, they are correct. The individual values were close to the true mean value in all replications in those cases.
The Conclusion section should be a separate paragraph with perspective point of view of the Authors and limits of the proposed work putting in evidence the end points and application of the findings in the food area.
Author’s response: Based on the reviewer’s suggestion, we have included a separate conclusion paragraph at the end the discussion section to explain authors perspective, limits of our work, end points, and applications in the food area.
Below is the separate conclusion paragraph added on page 17 in round 2 and revised again in round 3:
In conclusion, our work presented the evidence for low-level tolerance to one of the alternate antibiotics used for listeriosis treatment in some L. monocytogenes strains when exposed to sublethal doses of QAC, a most widely used biocide in the food industries and hospitals. Our data shows that some L. monocytogenes strains adapted to QAC may have a survival advantage in certain antibiotics during the early growth stages which is valuable for its risk assessment. Besides employing broth models using defined growth media, our study also used water samples that were spiked with varying known concentrations of QAC for better understanding of L. monocytogenes adaptation. As part of our future work, we will test water samples after sanitation for residual QAC levels and the presence of naturally surviving QAC-adapted and low-level antibiotic-tolerant L. monocytogenes strains in food residues. We will also explore the correlation between both homologous and heterologous tolerance in L. monocytogenes after QAC adaptation with respect to selected antibiotics.
Below are the new statements added in the above conclusion paragraph on page 17 in round 3:
Antibiotic resistance in foodborne isolates of L. monocytogenes is slowly increasing in many countries. While we did not find the clinical level of resistance to trimethoprim that may compromise in some cases to treat human listeriosis, however, the low-level tolerance to trimethoprim in QAC-adapted subpopulations may raise the possibility of future acquisition of resistance by L. monocytogenes. Therefore, our work may lead to early detection of low-level antibiotic tolerance in food isolates of L. monocytogenes induced by QAC-adaptation based on early growth dynamics. This information is useful for improving the sanitation steps and to eliminate safety concerns by preventing and mitigating the formation of low-level antibiotic tolerant strains if persisting in some food production and food processing environments.
Updates in the reference section made in round 2/3 on page 18 are shown below:
Aryal, M., Muriana, P.M., 2019. Efficacy of commercial sanitizers used in food processing facilities for inactivation of Listeria monocytogenes, E. coli O157:H7, and Salmonella biofilms. Foods 8 (12): 639. doi: 10.3390/foods8120639.
Centers for Disease Control and Prevention (CDC), 2021a. Listeria outbreak linked to fully cooked chicken. Food Safety Alert posted July 9, 2021. https://www.cdc.gov/listeria/outbreaks/precooked-chicken-07-21/index.html
Centers for Disease Control and Prevention (CDC), 2021b. Listeria outbreak linked to queso fresco made by El Abuelito Cheese Inc. Food Safety Alert posted May 14, 2021. https://www.cdc.gov/listeria/outbreaks/hispanic-soft-cheese-02-21/index.html
European Food Safety Authority and European Centre for Disease Prevention and Control (EFSA and ECDC), 2019. The European Union One Health 2018 Zoonoses Report. EFSA Journal 17(12):5926. DOI: https://doi.org/10.2903/j.efsa.2019.5926
Food and Drug Administration (FDA), 2021. Recalls, market withdrawals, & safety alerts. https://www.fda.gov/safety/recalls-market-withdrawals-safety-alerts (accessed July 25, 2021).
Track changes file of July 26, 2021
All revisions explained above in the manuscript round 2/3 are shown in blue in PDF track changes file of July 26, 2021.
In summary, this manuscript addresses one of the emerging research focus areas in food microbiology and food safety complex as part of the Foods Special Issue on "The Role of Food Chain in the Spread of Antimicrobial Resistance". We are thankful to the reviewer for the excellent suggestions on this manuscript. We are grateful for the critical reading and review.
Ramakrishna Nannapaneni, Corresponding Author
Submission Date: 26 July 2021
